# The Prognostic Significance of Early Glycemic Profile in Acute Ischemic Stroke Depends on Stroke Subtype

**DOI:** 10.3390/jcm12051794

**Published:** 2023-02-23

**Authors:** Paola Forti, Fabiola Maioli

**Affiliations:** 1Department of Medical and Surgical Sciences, Alma Mater Studiorum, University of Bologna, 40138 Bologna, Italy; 2Division of Internal Medicine, IRCCS Azienda Ospedaliero-Universitaria di Bologna, 40138 Bologna, Italy; 3Department of Integrated Health Care, Maggiore Hospital, 40133 Bologna, Italy

**Keywords:** acute ischemic stroke, lacunar, non-lacunar, blood glucose test, poor outcome

## Abstract

It is still unclear whether early glycemic profile after admission for acute ischemic stroke (IS) has the same prognostic significance in patients with lacunar and non-lacunar infarction. Data from 4011 IS patients admitted to a Stroke Unit (SU) were retrospectively analyzed. Lacunar IS was diagnosed by clinical criteria. A continuous indicator of early glycemic profile was calculated as the difference of fasting serum glucose (FSG) measured within 48 h after admission and random serum glucose (RSG) measured on admission. Logistic regression was used to estimate the association with a combined poor outcome defined as early neurological deterioration, severe stroke at SU discharge, or 1-month mortality. Among patients without hypoglycemia (RSG and FSG > 3.9 mmol/L), an increasing glycemic profile increased the likelihood of a poor outcome for non-lacunar (OR, 1.38, 95%CI, 1.24–1.52 in those without diabetes; 1.11, 95%CI, 1.05–1.18 in those with diabetes) but not for lacunar IS. Among patients without sustained or delayed hyperglycemia (FSG < 7.8 mmol/L), an increasing glycemic profile was unrelated to outcome for non-lacunar IS but decreased the likelihood of poor outcome for lacunar IS (OR, 0.63, 95%CI, 0.41–0.98). Early glycemic profile after acute IS has a different prognostic significance in non-lacunar and lacunar patients.

## 1. Introduction

A transitory hyperglycemia is frequent in the earliest phase of ischemic stroke (IS) as a part of the acute stress reaction [1]. In IS patients, admission hyperglycemia has a robust association with infarct size, initial clinical severity, neurological worsening, and short-term outcome [1,2]. Although a causal association is still unproven, some evidence suggests that hyperglycemia can worsen brain damage by fueling inflammatory mechanisms in the hypoperfused, but still potentially salvageable, area around the ischemic core (penumbra), thus exacerbating reperfusion injury [1].

Lacunar stroke is thought to have a more favorable outcome than other IS subtypes because of its very low acute mortality, but up to 30% of lacunar patients actually develop early neurological deterioration that often leads to important disability [3,4]. Lacunar stroke, being typically caused by the occlusion of small perforating end arteries, usually lacks an ischemic penumbra [4,5]. Therefore, it has been hypothesized that admission hyperglycemia might be unrelated to short-term outcome of lacunar IS or even be beneficial by providing readily available fuel for the viable brain tissue around the ischemic core [6,7]. However, available clinical cohort studies of this issue are few, inconsistent, and based on a single glucose measurement [8,9,10,11]. In acute IS patients without diabetes, stroke onset is usually accompanied by a moderate glycemia increase (<10 mmol/L) followed by a physiological decrease within the subsequent 24 h [12]. However, persistent and delayed hyperglycemia within the first 48 h after stroke can also occur, and these glycemic trajectories seem to have a stronger association with poor short-term outcomes of IS than a single glycemic measure on admission [13,14].

Spontaneous hypoglycemia is also a known predictor of adverse short-term outcomes in both critical [15] and non-critical [16] inpatients but has received small attention by stroke researchers, because it is thought to be a rare occurrence on admission for acute IS [17]. In particular, it is unknown whether its prognostic significance is affected by IS subtype.

In this retrospective cohort study, we investigated whether IS subtype affects the association between early glycemic profile and short-term stroke outcome.

## 2. Materials and Methods

This is a retrospective, observational single-center study based on a cohort of 4052 IS patients aged ≥18 years who, between January 2006 and December 2018, were consecutively admitted to the Emergency Department of the Maggiore Hospital (Bologna, Italy) within 24 h after symptom onset and subsequently transferred to the local Stroke Unit (SU). At SU admission, written informed consent for future research use of all data included in their medical records was sought from patients or their legally authorized representatives. The study was conducted in accordance with the Declaration of Helsinki. The Maggiore Hospital Ethics Committee approved the study (approval number CE16092). The dataset is not publicly available due to privacy reasons, but data in anonymized form are available on request from the corresponding author.

All patients had at least one CT-head scan at hospital admission. IS was classified as lacunar if presenting with one of the five classic lacunar syndromes (pure motor hemiparesis, sensorimotor stroke, ataxic hemiparesis, pure sensory syndrome, and dysarthria-clumsy hand syndrome) [18]. Neuroimaging evidence of lacunar infarction was defined as a subcortical brain infarct <15 mm [4] on CT or MRI. Non-lacunar IS included large artery atherosclerosis, cardioembolic, and other or unknown etiologies as determined at the time of the patient’s discharge [19]. Patients were treated according to standard guidelines for management of acute stroke, which recommend avoidance of intravenous fluids containing glucose and set the glucose target for glycemic control between 7.8 and 10.0 mmol/L [20]. Information about demographic, medical characteristics, and stroke outcome was obtained from medical records.

Random serum glucose (RSG) was measured at the Emergency Department, usually within 3 h of admission. Fasting serum glucose (FSG) and glycated hemoglobin ((HbA1c), Diabetes Control and Complications Trial aligned results) were determined the morning after SU admission (median time after admission, 24 h; range, 12–48 h) as a part of the routine biochemistry tests performed on venous blood samples drawn after an overnight fast. All measurements were performed using automated methods at the same central laboratory. Hyperglycemia was defined as serum glucose ≥ 7.8 mmol/L (140 mg/dL), corresponding to the standard threshold for treatment in acute stroke [20]. Hypoglycemia was defined as serum glucose < 3.9 mmol/L (70 mg/dL), corresponding to the standard glucose alert value in clinical practice [21].

Diabetes was defined as pre-admission diagnosis (self-report, evidence from available medical records, or use of antidiabetic agents), new diagnosis made during SU stay, or retrospective diagnosis based on admission HbA1c ≥6.5% (48 mmol/mol) [22]. Although not fully concordant with traditional blood glucose criteria [23], HbA1c is a convenient choice for diabetes screening in acute stroke because, differently from blood glucose, it remains unaffected by stress response [24].

Prestroke disability was defined as admission modified Rankin Scale ≥ 2 [25]. Initial stroke severity was assessed using the National Institutes of Health Stroke Scale (NIHSS) measured at ED arrival [26]. Acute reperfusion treatment on admission was also recorded.

We defined a composite poor outcome including early neurological deterioration (increase in NIHSS score ≥ 4 points or death within 24 h after admission) [27], very severe stroke at SU discharge (NIHSS score ≥25 [28]), or death within 30 days after stroke onset as ascertained from the Italian Regional Mortality Registry.

Variables were reported as median (25th–75th percentile) or number (percentage). Univariate associations were tested using the Mann–Whitney or chi-square test as appropriate. In preliminary univariate analyses, patients were categorized into five glycemic trajectories based on RSG and FSG values: hypoglycemia (low FSG or FSG), persistent normoglycemia (normal RSG and normal FSG); persistent hyperglycemia (high RSG and high FSG), delayed hyperglycemia (normal RSG and high FSG), and decreasing hyperglycemia (high RSG and normal FSG). Univariate analyses suggested the possibility of non-linear associations but also showed that for some glycemic trajectories, lacunar patients were so few as to preclude any subsequent reliable statistical estimation by multivariable models based on the multiple categorization of predictors or complex non-linear functions. Therefore, multivariable analysis was performed by a logistic regression model testing for the interaction of IS subtype with both admission RSG and a continuous indicator of early glycemic profile calculated as the difference of FSG and RSG. We assumed that a positive difference in this value indicated a trend toward increasing glucose values, whereas a negative difference indicated a trend toward decreasing glucose values. The model also tested for the interaction of continuous glucose measurements with diabetes, which is known to weaken the adverse prognostic impact of stress hyperglycemia in acute stroke [13]. Covariates included age, sex, disability, reperfusion therapy, and admission NIHSS (log-transformed). Based on results from univariate analyses, this logistic regression model was applied to two partially overlapping subsets of subjects. The first subset included only patients with RSG and FSG > 3.9 mmol in order to focus on the association of poor outcome with high admission RSG and increasing early glycemic profile in the absence of hypoglycemia. The second subset included only patients with FSG <7.8 mmol/L in order to test whether high admission RSG or increasing glycemic profile could take a favorable prognostic significance in the absence of persistent or delayed hyperglycemia. Corollary analyses did not evidence interactions of reperfusion therapy with either admission RSG or early glycemic profile.

Analyses were performed with R software version 3.5.3 and Harrell’s rms package [29]. Significance for *p* value was set at 0.050 (two-tailed). The study power was 0.80 for an OR of 1.3.

## 3. Results

### 3.1. Characteristics of Patients

The final cohort included 4011 patients (age range 19 to 101 years). We excluded 22 patients who refused/were unable to provide informed consent, nine patients with missing clinical or laboratory data, and 10 patients lost at follow-up after SU discharge. Excluded patients did not differ by age, sex, and admission NIHSS. Reperfusion therapy was administered to 678 patients (560 intravenous fibrinolysis, 29 mechanical thrombectomy, and 89 both). A lacunar stroke syndrome was clinically diagnosed in 834 patients, the most frequent presentation being pure motor (61%), followed by ataxic (16%), and sensory-motor (9%). Only 269 (32.2%) of these patients had evidence of lacunar infarction at the CT scan routinely performed at day 3 after admission. Of the 565 patients with negative CT scan, only 129 also underwent MRI (based on clinical judgment of the attending physician), but lacunar infarction was confirmed in 119 of them (92.2%).

The most frequent etiology for non-lacunar IS (*n* = 3177) was cryptogenetic (44.2%), which was followed by cardioembolic (40.3%) and large artery atherosclerosis (12.5%). Diabetes was diagnosed in 1161 patients (844 known at admission, 88 newly diagnosed during SU stay, and 229 retrospectively diagnosed based on HbA1c). Table 1 compares baseline characteristics by IS subtype.

Lacunar patients were more likely to be younger, men and without prestroke disability than non-lacunar patients. They were also less likely to present with high NIHSS and to undergo reperfusion treatment. Lacunar patients had lower RSG and higher prevalence of hypoglycemia and persistent normoglycemia than non-lacunar patients. However, continuous early glycemic profile did not differ by stroke subtype. The overall prevalence of hypoglycemia in the cohort was 7.8%, with only 13 cases due to low admission RSG. Serious hypoglycemia (<3 mmol/L) [21] was rare (less than 1%). As expected, single poor outcomes as well as composite poor outcome were more likely in non-lacunar IS, with early neurological deterioration being the most common occurrence.

### 3.2. Univariate Association of Glucose Profile Categories with Poor Outcome

Table 2 show the distribution of combined poor outcome across different glycemic trajectories by IS subtype. In non-lacunar patients, poor outcome appeared markedly associated with hyperglycemic trajectories, delayed hyperglycemia having the highest proportion of cases. In lacunar patients, conversely, the highest proportion of cases was found for hypoglycemia, but numerosity was too small for a reliable statistical test.

### 3.3. Multivariable Analysis

When applied to the 3699 patients without hypoglycemia, multivariable logistic analysis for prediction of poor outcome showed that IS subtype had a significant interaction with both admission RSG (*p*-value = 0.032) and early glycemic profile (*p*-value = 0.028). Significant interactions were also found between diabetes and both glucose predictors (*p*-value < 0.001). Figure 1 shows how higher admission RSG and increasing glycemic profile were both associated with higher likelihood of poor outcome in non-lacunar but not in lacunar IS. In non-lacunar IS, the adverse prognostic significance of both predictors was weaker if the patient had diabetes. When applied to the 3539 patients without persistent or delayed hyperglycemia, multivariable logistic analysis showed significant interactions of IS subtype with both admission RSG (*p*-value 0.004) and glycemic profile (*p*-value 0.003); no significant interactions were found with diabetes.

Figure 2 shows how, in non-lacunar IS, both higher admission RSG and increasing glycemic profile remained associated with poor outcome. In lacunar IS, conversely, admission RSG was no more a statistically significant predictor, and patients with an increasing glycemic profile even had a lower likelihood of poor outcome.

In corollary analyses contrasting patients with hypoglycemia against the rest of the study cohort, there was no significant association with poor outcome and no significant interaction with IS subtype or diabetes (*p*-value > 0.200 for all).

## 4. Discussion

This study shows that admission RSG and early glycemic profile do not have a similar prognostic significance in patients with lacunar and non-lacunar IS. Higher admission RSG and increasing glycemic profile were both associated with a composite poor outcome in non-lacunar IS, even if the associations were weakened in patients with diabetes. No similar associations were found in lacunar patients, among whom an increasing glycemic profile might even take a favorable prognostic significance when blood glucose remained below the threshold for hyperglycemia.

According to findings from animal models and human cohorts [1,7], the adverse association of stress hyperglycemia with severity and outcome of IS depends on the presence of an ischemic penumbra around the irreversibly injured tissue of the infarct core. From an evolutionary perspective, stress hyperglycemia is an adaptive response that aims to provide the brain with ready fuel during an acute threat [30]. In the ischemic penumbra of a brain infarction, however, glucose excess would favor anaerobic glycolysis, so promoting harmful intra- and extra-cellular processes (acidosis from release of lactic acid, accumulation of glutamate, oxygen radicals production) that can accelerate the transition to irreversible injury, expand infarct size, and increase the risk of hemorrhagic transformation [1,2,6,31]. Collateral circulation is a critical determinant of cerebral perfusion in acute cerebral ischemia and may also be important in maintaining perfusion to penumbral regions [32]. A study of 309 IS patients undergoing endovascular thrombectomy clearly showed how higher blood glucose levels significantly reduced the likelihood of good outcome in patients with good collaterals but not in those with poor collaterals [33].

Lacunar infarction usually lacks an ischemic penumbra because this IS subtype generally results from the occlusion of small cerebral arteries with poor collateral circulation [5]. Therefore, in lacunar IS, stress hyperglycemia has been hypothesized to be indifferent to stroke outcome or even to have a favorable effect by acting as a fuel source for the vital brain cells around the ischemic core [7]. However, available literature on this subject is conflicting and limited to few studies, all of which used a single admission RSG measurement and a composite outcome of death or functional disability. In 635 patients from the placebo arm of the TOAST trial [8], those with lacunar IS and higher glycemia had a better 3-month outcome than their normoglycemic counterparts, whereas the opposite occurred in those with non-lacunar IS. A similar finding was reported in 1375 IS patients from two clinical trials of lubeluzole [11]. Conversely, two prospective studies, of 1012 [10] and 2020 patients [9], respectively, reported no association of admission hyperglycemia with poor outcome in lacunar IS. A comparison of these studies is difficult, because only the first one used admission RSG as a continuous variable [8], while the others dichotomized it at different cutoffs (from 6.1 [9] to 8 mmol/L [10,11]).

Our study confirmed that an increasing glycemic profile in the early phase of acute IS was unrelated to the short-term outcome of lacunar patients without hypoglycemia. However, our study also showed that an increasing glycemic profile was associated with lower risk of poor outcome in lacunar patients who did not develop sustained or delayed hyperglycemia during the earliest phase of admission. A possible explanation is that this subgroup of lacunar patients includes those with a smaller and more stable infarction. However, some lacunar IS actually have a penumbra area [4]. Moreover, even when well perfused, the brain tissue surrounding an infarct core is exposed to inflammatory damage, excitotoxicity, and a spreading depolarization wave that can affect blood flow and promote enlargement of the infarction [34]. These mechanisms might be exacerbated by an increased glucose availability [35,36]. Since lacunar IS is small by definition, these events may have no clinical relevance unless blood glucose reaches consistently high levels. This might explain the apparent protective effect of an increasing glycemic profile observed in our lacunar patients whose blood levels remained below the threshold for hyperglycemia. These patients might just be the ones with the better trade-off between the pros and cons of an increased glucose availability in their brain. The finding that suggests this favorable association, however, should be considered with caution, because we performed multiple statistical analyses on overlapping subsets of the same population, so increasing the possibility of obtaining statistically significant results by chance alone.

A noticeable corollary finding of this study is the lack of association between hypoglycemia and poor outcome regardless of IS subtype and other confounders. Mechanisms by which hypoglycemia could worsen acute ischemic brain damage include: direct neurodamage by bioenergetic deficit; stress of the cardiovascular system as a consequence of the autonomic response; pro-inflammatory and pro-thrombotic effects worsening ischemic damage; and alterations of cerebral vasoregulation [1,31,37]. Stroke literature mostly focused on iatrogenic hypoglycemia occurring during trials of tight glucose control [38]. In these trials, hypoglycemia was usually unrelated to stroke clinical outcomes [39,40,41,42]. The only exception is the GIST-UK trial [43], reporting higher mortality among the insulin-treated patients with the greatest blood glucose reductions. Experimental data from animal stroke models are few and conflicting [2,44]. Data about spontaneous hypoglycemia on admission in IS cohort studies are also limited [45,46,47,48,49]. Reported occurrence of this issue varies from zero [45] to 49% [47]. Evidence of its association with higher risk of death or poor functional status at 3 months is similarly unclear [46,48]. Although there is consistent evidence that spontaneous hypoglycemia is a mortality predictor in patients hospitalized because of several acute conditions, the existence of a true causal relationship is still debated [15,16]. Studies of older inpatients suggest that spontaneous admission hypoglycemia might be just an epiphenomenon of poor pre-admission health [50].

The strengths of our study include the large number of participants, the use of two subsequent glucose measures to define an indicator of early glycemic profile, and the attention paid to identifying diabetes, which might confer a higher cellular tolerance to acute hyperglycemia [15].

Our study has also several major limitations. First, about one-half of the patients with lacunar syndromes lacked neuroradiologic confirmation, and an MRI scan was not systematically performed. Although classic lacunar syndromes are highly suggestive of small deep cerebral infarction, the correlation between clinical syndromes, neuroradiological features and histological findings is not absolute [4], and different etiologies can be found in about 17% of cases [51,52]. However, the radiologic confirmation of lacunar IS can be problematic in clinical practice: TC and conventional MRI often lack the necessary resolution to detect lacunar infarctions, while systematic recourse to techniques with superior precision, such as diffusion-weighted MRI, is not routinely feasible in clinical settings [53]. Second, our measure of glycemic profile was very approximative because it was based only on two, not fully comparable, glucose measurements. Third, since we performed multiple analyses on overlapping subsets of the same population, we cannot exclude that some of our statistically significant results were actually due to chance. Fourth, the study retrospective design prevents from drawing causal inferences. Fifth, although analyses were adjusted for admission clinical severity, no direct measure of infarction size was available. Information about occurrence of epileptic seizures would also have been of interest, as this disorder can be exacerbated under conditions of hyper- or hypoglycemia. Unfortunately, however, such information was not available. Finally, our database did not include information about a major clinical outcome such as post-discharge disability.

In conclusion, this study shows that early glycemic profile after acute IS has a different prognostic significance in lacunar and non-lacunar patients. An increasing glycemic profile is clearly associated with poor short-term outcome in non-lacunar patients, but the unfavorable association is not evident in lacunar patients. So far, clinical trials have failed to identify clinical benefits from intensive glucose control by insulin in acute stroke inpatients [1,2,31]. However, other antidiabetic agents, such as GLP-1 receptor agonists and DPP-4 inhibitors, are under consideration [1]. Our results suggest that future trials of glucose-lowering agents in acute IS should take into account the possibility of different management standards for lacunar and non-lacunar patients.

## Figures and Tables

**Figure 1 jcm-12-01794-f001:**
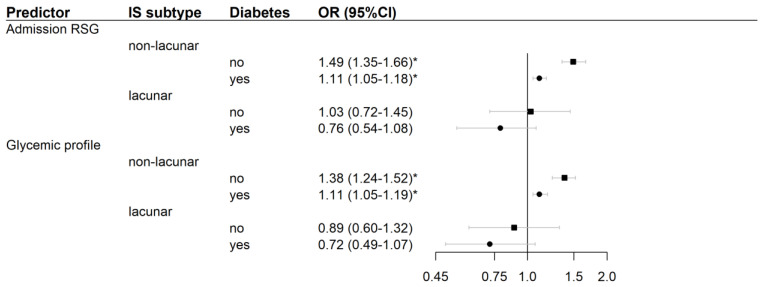
Association of continuous glucose predictors with poor outcome of ischemic stroke (IS) in patients without hypoglycemia. Odds ratios (OR) are for a unit increase in random serum glucose (RSG) on admission and early glucose profile (defined as the difference of fasting serum glucose measured the morning after admission and admission RSG). The logistic model was adjusted for age, sex, prestroke disability, admission National Institutes of Health Stroke Scale score, and reperfusion therapy. Patients were stratified by IS subtype, and diabetes status because the logistic model included significant interactions of RSG and early glycemic profile with both these characteristics. Poor outcome was defined as early neurological deterioration, severe stroke at Stroke Unit discharge, or death at 30 days after stroke onset. Analysis included only patients with RSG and FSG ≥ 3.9 mmol/L. Data are for 2088 non-lacunar patients without diabetes (400 cases), 862 non-lacunar patients with diabetes (205 cases), 508 lacunar patients without diabetes (23 cases), and 241 lacunar patients with diabetes (3 cases). Significant ORs (*p*-value < 0.05) are marked with *.

**Figure 2 jcm-12-01794-f002:**
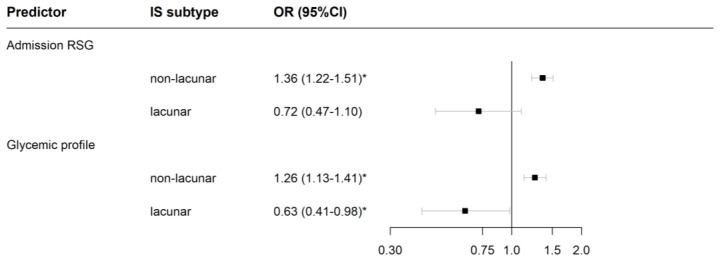
Association of continuous glucose predictors with poor outcome of ischemic stroke (IS) in patients without persistent or delayed hyperglycemia. Odds ratios (OR) are for a unit increase in random serum glucose (RSG) on admission and early glucose profile (defined as the difference of fasting serum glucose measured the morning after admission and RSG). The logistic model was adjusted for age, sex, prestroke disability, admission National Institutes of Health Stroke Scale score, and reperfusion therapy. Patients were stratified by IS subtype because the model included significant interactions of admission RSG and early glycemic profile with this characteristic. Poor outcome was defined as early neurological deterioration, severe stroke at Stroke Unit discharge, or death at 30 days after stroke onset. Analysis included only patients with FSG < 7.8 mmol/L. Data are for 2781 non-lacunar patients (501 cases) and 758 lacunar patients (31 cases). Significant ORs (*p*-value < 0.05) are marked with *.

**Table 1 jcm-12-01794-t001:** Baseline characteristics and outcomes by ischemic stroke subtype.

Characteristics	Non-Lacunar (*n* = 3177)	Lacunar (*n* = 834)	*p* Value
Demographic and clinical data			
Age, years	80 (71–86)	76 (67–84)	<0.001
Male sex	1426 (45.8)	471 (56.5)	<0.001
Prestroke disability	987 (31.1)	223 (26.7)	0.015
Diabetes	901 (28.4)	260 (31.2)	0.111
Admission NIHSS score	8 (3–17)	3 (2–5)	<0.001
Reperfusion therapy	595 (18.7)	83 (10.0)	< 0.001
Glucose measurements			
Admission RSG, mmol/L	6.7 (5.7–8.4)	6.2 (5.4–7.7)	<0.001
FSG, mmol/L	5.2 (4.5–6.4)	4.9 (4.3–5.6)	<0.001
Early glycemic profile, mmol/L	−1.4 (−0.5–−2.6)	−1.4 (−0.6–−2.4)	0.697
Early glycemic profile categories			<0.001
Hypoglycemia	227 (7.2)	85 (10.2)	
Persistent normoglycemia	1901 (59.8)	545 (65.4)	
Decreasing hyperglycemia	654 (20.6)	128 (15.3)	
Persistent hyperglycemia	318 (10.0)	65 (7.8)	
Delayed hyperglycemia	77 (2.4)	11 (1.3)	
Outcome			
Early neurological deterioration	201 (6.3)	19 (2.3)	<0.001
NIHSS > 24 at Stroke Unit discharge	92 (2.9)	1 (0.1)	0.001
1-month mortality	427 (14.9)	14 (1.7)	<0.001
Composite poor outcome	644 (20.3)	32 (3.8)	<0.001

Data are reported as n (%) or median and (25th–75th percentile). *p* values are for Mann–Whitney or chi-square test. To convert mmol/L to mg/dL, multiply by 18.018. Glycemic profile was defined as the difference of FSG and RSG. Abbreviations: FSG, fasting serum glucose; NIHSS, National Institutes of Health Stroke Scale; RSG, random serum glucose.

**Table 2 jcm-12-01794-t002:** Univariate association of categorical early glycemic trajectories with poor outcome by stroke subtype.

	Hypoglycemia	Persistent Normoglycemia	Decreasing Hyperglycemia	Persistent Hyperglycemia	Delayed Hyperglycemia	*p*-Value
**Non-lacunar**	39/227 (17.2)	309/1901 (16.3)	153/654 (23.4)	105/318 (33.0)	38/77 (49.4)	<0.001
**Lacunar**	6/85 (7.1)	19/545 (3.5)	6/128 (4.7)	1/65 (1.5)	0/11 (0.0)	0.380

## Data Availability

The dataset is not publicly available to privacy reasons, but data in anonymized form are available on request from the corresponding author.

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
