# Peer review of "The Prognostic Significance of Early Glycemic Profile in Acute Ischemic Stroke Depends on Stroke Subtype"

_jcm, 2023, doi:10.3390/jcm12051794_

Round 1
Reviewer 1 Report
In this interesting study, the authors performed a retrospectively analyses to evaluate whether early glycemic profile after admission for acute ischemic stroke (IS) has the same prognostic meaning for patients with lacunar and non-lacunar subtypes.
However, there are several questions/concerns as follows:
1. Patients who present with symptoms of a lacunar stroke may be described as having lacunar stroke syndrome (LACS), but in the study the Lacunar Stroke was diagnosed only by clinical criteria without radiologic imaging which may cause much errors. And most importantly, the diagnosis of lacunar and non-lacunar stroke subtypes was the basis of the study.
2. In addition, one of the measures of the poor outcome in the research was 1-month mortality. Maybe the Modified Rankin Scale (mRS) measuring degree of disability/dependence after a stroke seems more suitable.
3. The English writing is poor. It is better to revise the manuscript by people whose native language is English. (eg: It still unclear whether early glycemic profile after admission for acute ischemic stroke (IS) has the same prognostic meaning for patients with lacunar and non-lacunar subtype. )
Author Response
Thank you very much for your kind comments.
We completely agree with the concerns raised by Reviewer 1 in point 1 and 2. Systematic radiologic confirmation for lacunar infarction and data about functional status at 1-month would have greatly increased the quality of our paper. Unfortunately, as it often happens with retrospective data obtained from clinical practice rather than prospective trials, we lack the information needed to practically address these issues. However, we had already mentioned them when discussing the limitations of the study and, in the revised version, we attempted to emphasize them further (see lines 295-303 and lines 312-313).
The reviewer was also absolutely right about the poor quality of writing (point 3). We addressed this issue with the help of a native English speaking colleague, now mentioned in the Acknowledgements. The manuscript (including the title) underwent an extensive revision for English language and grammar. We hope that now the text is more pleasant to read and easier to understand.
Reviewer 2 Report
This manuscript aims to evaluate whether IS subtype affects the association between early glycemic profile and short-term stroke outcome.
The title is adequate and reflect the objective of the manuscript. The abstract, keywords, materials and methods, results discussion, conclusions, and interpretations reflect the outcomes of the manuscript. The figures and tables are adequate and support the results achieved.
It's nice to see that the authors clearly point out the strengths and the several major limitations of the retrospective cohort study conducted.
In conclusion, the presented manuscript is interesting and may be applicable in the routine practice of healthcare professionals.
Author Response
Thank you very much for you very kind comments!
(No concerns raised by reviewer 2)
Please notice that the manuscript (including the title) underwent an extensive revision for English language and grammar. We hope that now the text is more pleasant to read and easier to understand.
Reviewer 3 Report
In this manuscript, Paola Forti and Fabiola Maioli report a clinical study to investigate the possible relationship between early glycemic profile and functional outcome in patients with lacunar and non-lacunar subtype. This is an observational, not mechanistic, but interesting study with a large collection of patients that brings more knowledge on the relationship between glucose at admission and post-stroke functional outcome.
The authors discuss the controversial aspects of analogous clinical studies. I believe that there can be no controversy when clinical studies are performed in different ways or they are not completely equivalent. Unfortunately, main bottlenecks on clinical research are always lack of standardization and incomplete data of patients; to give an example in this case, lack of neuroradiologic confirmation of lacunar infarction.
I have three main comments (all of them for discussion):
1) After reading the discussion, it seems that the authors relate the levels of early glycemy with the magnitude of cerebral blood perfusion in the penumbra to justify the shot-term functional outcome. The authors state in several parts of the manuscript that lacunar strokes lack of ischemic penumbra. They use this argument to justify the hypothesis that early variations in glycemia does not affect short-term outcome of lacunar stroke due a complete lack of blood flow.
It is widely accepted that the brain tissue surrounding the infarct core is affected for the inflammatory response, toxicity mediated by glutamate and Ca2+ overload, reactive oxygen species and spreading depolarization waves (which affects blood flow in the penumbra, see for example (https://doi.org/10.3390/ijms23137449). However, it is difficult to reconcile that, even in the absence of collateralization in lacunar strokes, the well-perfused surrounding brain tissue can not experience events of inflammation or neurotoxicity that can contribute to enlarge the infarct and affect functional outcome. In other words, are the effects of early glucose levels exclusively linked with the level of cerebral blood perfusion in penumbra? if there is not blood flow (i.e. lacunar stroke/absence of collaterals) do the glucose levels have any influence on infarct evolution and functional outcome due inflammation or neurotoxicity in the well-perfused brain tissue? (see for example the effect of hyperglycemia on massive neutrophil infiltration https://doi.org/10.1016/S0304-3940(99)00889-7); or Neurotoxicity influenced by glucose; https://doi.org/10.2174/1567205014666170117104053).
Accordingly with the author hypothesis in lacunar strokes, perhaps the authors want also to check and mention this study (https://doi.org/10.1161/STROKEAHA.118.022167)
2) In relation with the previous point, how the efficiency of reperfusion (endovascular therapy) did affect functional outcome dependently of glycemic profile?
3) Epileptic seizures can be exacerbated under conditions of hyper- or hypoglycaemia. A significant fraction of stroke patients (highly variable between studies) show seizures during the hyperacute stroke phase. Do the authors consider these aspects on this study?
Author Response
Thank you very much for your kind and useful comments.
The concerns raised by reviewer 3 were addressed as follows:
1) The reviewer suggested to take into account the possibility that excess blood glucose can cause damage in the brain area surrounding the ischemic core even in absence of ischemic penumbra. We are indebted to the reviewer for this suggestion and the useful references (all included in the revised version of the manuscript). The issue is discussed in lines 224 to 271 of the revised version.
2) About a possible interaction of glycemic profile with reperfusion therapy, we tested for this interaction and verified that it is not significant in our population (see lines 123-125).
3) In lines 1199-1201 of the discussion we acknowledge that information about epileptic seizures and their relationship with glycemic profile would have increased the quality of the manuscript. Unfortunately, as we made it clear in the text, such information is unavailable so we can only mention the issue in the discussion.
Please notice that the manuscript (including the title) underwent an extensive revision for English language and grammar. We hope that now the text is more pleasant to read and easier to understand.
Reviewer 4 Report
Congratulation!
The retrospective cohort study investigated by authors proved that IS subtype affects the association between early glycemic profile and short-term stroke outcome. The study was carried out well.
Figure 1. Suggestions OR(95%CI) column have to be move to left in order to increase the quality of figure. Ad a p value column or mark with * where the p value is under 0.05 (rows 1, 2 and 5, 6).
Figure 2. - the same suggestions with figure 1.
Author Response
Thank you very much for your kind comments.
We amended both figures according to your suggestions (OR column was moved to the left and a mark * was added to significant values).
Please notice that the manuscript (including the title) underwent an extensive revision for English language and grammar. We hope that now the text is more pleasant to read and easier to understand.
Round 2
Reviewer 1 Report
The revised manuscript is much better than previous version. Thanks for your serious revision.
Reviewer 3 Report
The authors have adequately revised their manuscript, which has improved the paper.